# Advising parents when their child has a fever: a phenomenographic analysis of nurses' perceptions when working at a telephone helpline, at primary care or at a paediatric emergency department in Sweden

Emma Westin [ID] ,[1,2] Ingrid L Gustafsson [ID] ,[1,3] Anders Svensson [ID] ,[1,4] Märta Sund-Levander [ID] ,[5] Carina Elmqvist [ID] [1,6]

For numbered affiliations see end of article.

**Correspondence to**
Emma Westin;
emma.westin@lnu.se

## ABSTRACT

**Objectives** To describe nurses' perceptions of advising parents when their child has a fever.

**Design/method** Inductive, descriptive study with a qualitative, phenomenographic approach.

**Participants and setting** A purposive sampling was used. To be included, the 24 online interviewed nurses had to have experience advising parents of febrile children between birth and 5 years of age. They were recruited from three different parts of the healthcare system from four regions in the south of Sweden.

**Results** The nurses described advising parents when their child has a fever as four different kinds of balancing acts: balancing between the parents' story and objective assessment, balancing between listening and teaching, balancing between self-confidence and trust in the expert, and balancing between independence and having someone by one's side.

**Conclusions** Giving advice to parents when their child has a fever is a process where the nurse needs to listen, assess and give advice based on the situation. This requires a correct assessment that depends on the parents' story. Creating a trusting relationship is perceived as necessary for parents to assimilate the advice that is provided. What dominates are the nurses' perceptions of the inner qualities required to achieve a balance in the process, for example, the importance of experience and security in their professional role, while it is also necessary to get support from colleagues.

## INTRODUCTION

In Sweden, parents can generally seek advice via telephone nursing services or within primary care.[1] Advising is defined as helping the parents assess the need for healthcare, give advice and information about measures that the parents can do themselves, and, if necessary, refer to appropriate level of care.[2] The telephone helpline 1177 Vårdguiden (henceforth

1177) is accessible around the clock all year round. The main focus for the tele-nurses at 1177 is to advise on different health problems.[3] To support their advice, they use a computerised decision support tool (CDST); the same system is also frequently used by nurses who work in primary care. Primary care centres are responsible for preventing, diagnosing and treating illnesses, and, if necessary, sending referrals to specialist care.[4] They must offer services for both planned and unplanned visits in general medicine during the day, as well as some on-call services. Primary care also includes child healthcare.[5] The goal of child healthcare is to promote health and prevent illness in children from birth until the child starts school by following the child's development and health status and providing support to parents.[6] If the child has acute symptoms, the parents should go directly to the paediatric emergency department, whose main purpose is to provide fast and highly specialised care for acute,

sometimes life-threatening conditions. However, it is common for parents to go there for minor illnesses; in those cases, advice from a nurse may sometimes be the only course of action.[7]

Feverish children cause concerns and are among the most common reasons why parents contact telephone nursing services,[8–10] primary care[11 12] or paediatric emergency departments.[7 12–15] The most common expectations parents have is to get their child a physical examination by a physician, and to receive reassurance and information about warning signs.[16] The initial meeting within the healthcare system is usually with a nurse, who is responsible both for creating a trusting relationship and for providing adequate information.[17] According to Halldórsdóttir,[18] this first encounter is decisive in whether the meeting and the relationship will be perceived as caring or non-caring.[18] It is important that the nurses can identify the parents' needs, which can vary depending on background, expectations and previous experience, in order to decide what strategies that can be used to empower them.[19] Previous studies have examined parents' views of having a child with fever, with or without contact with the healthcare system.[20 21] It is also recognised that nurses can play an important role in providing knowledge and advice to these parents.[19] However, previous research also shows that advising can be experienced as an advanced and demanding task.[22] Knowledge about nurses' perceptions is needed in order to prepare and support nurses taking on this role. This study, therefore, aims to describe nurses' perceptions of advising parents when their child has a fever, in a Swedish context.

## METHODS

### Design

An inductive, descriptive study with a qualitative approach was conducted, that is, phenomenography which describes the different perceptions that exist of a phenomenon. This is referred by Marton[23] as the second-order perspective. The first-order perspective concerns objective observations, that is, facts.[23] Perceptions, on the other hand, are unspoken, implied and unreflected. It is perceptions that form the frame of reference for our thinking and the basis for our opinions.[24]

### Participants

Purposive sampling was used to obtain a rich and varied understanding of the phenomenon. That included nurses from four different regions in southern Sweden from three different parts of the healthcare system; the telephone helpline 1177 Vårdguiden; primary healthcare, including child healthcare and the paediatric emergency department. To be included, the nurses had to have experience advising parents of febrile children between birth and 5 years of age. The first author contacted the department head at each workplace, who then sent out information about the study to the nurses. Those who then chose to participate contacted the author directly. The participants received both written and oral information about the study and provided informed written consent. They were informed that participation was voluntary and that they had the right to cancel their participation in the study at any time. To find all the different perceptions that can exist in a group, Larsson och Holmström[25] believe that approximately 20 interviews are sufficient, therefore, the goal was to get between 6 and 8 participants from each part of the healthcare system and with a geographical spread. This resulted in a total of 24 participants. The participants varied between 27 and 64 years of age (average 45), and their experience varied from 11 months to 25 years. Of the 24 nurses, 13 had at least one specialist nurse education, most commonly paediatric nurse or primary healthcare nurse (see table 1).

### Data collection

Data were collected through semistructured interviews. The interviews were conducted via Zoom by the first author during the period from March 2021 to October 2021. The interviews all began with the same open-ended question: 'What is advising parents whose child has a fever for you?', followed by a number of predefined questions according to an interview guide (see online supplemental file 1). Follow-up questions such as 'Can you tell me more?' and 'What do you mean when you say…?' were used in order for the participants to reflect on the phenomenon and deepen their reasoning. Two pilot interviews were conducted, and since they did not lead to any changes in the interview guide, both were included in the study. The interviews lasted between 22 and 51 min (average 37 min) and were transcribed verbatim. Only the person who conducted the interviews had access to the recorded interviews, that is, the first author. While

| Table 1 | Demographics of the 24 nurses | | |
|---|---|---|---|
| | Telephone helpline 1177 (n=8) | Primary healthcare central and child health central (n=8) | Paediatric emergency department (n=8) |
| Gender (F/M) | 8/0 | 8/0 | 6/2 |
| Age (average years) | 48–64 (55.75) | 34–64 (46.38) | 27–44 (34.25) |
| Specialist nurse education (yes/no) | 2/6 | 7/1 | 4/4 |
| Work experience (average years) | 0.9–15 (5.6) | 3.5–25 (8.6) | 2.5–10 (7.8) |

## Box 1 The seven steps of data analysis, according to Larsson and Holmström[25]

1. The text of the transcribed interviews was read through repeatedly.
2. The answers to the interview-questions were marked. This second step was done again and again to make sure that the marked text correlated to the questions.
3. In the third step, the authors looked in the marked text for what the focus of the nurse's attention was and how she/he described her/his way of advising parents when their child has a fever. This resulted in a preliminary description of each nurse's predominant way of understanding their work, in other words, their perceptions.
4. The descriptions were grouped into categories based on similarities and differences. This step was done and redone several times, like a journey between parts and whole to ensure that the categories were found in the material and vice versa.
5. Non-dominant ways of understanding were searched out. For an overview of dominant (++) and non-dominant (+) ways of understanding nurses' perceptions of advising parents when their child has a fever (see table 2).
6. The hierarchical way the categories were related to one another was defined, structuring the outcome space.
7. Based on the outcome space, each category of description was assigned a metaphor, in this case, the categories came out as four different kinds of balancing acts.

the interviews were transcribed, they were deidentified from anything that could reveal the participant's identity. Only those who participated in the analysis had access to the transcriptions.

### Data analysis
Data analysis was performed following the seven-step model of Larsson and Holmström[25] (see box 1). The analysis is not linear, but rather a back-and-forth motion between the whole and the parts. The analysis was mainly performed by the first, second and last author, but all authors participated and discussed the various steps during the process. The first, second and last authors started out each step in the analysis together, then the first author finished each step but the second and last authors reflected on and verified the outcome. Regularly scheduled meetings were done with the whole group.

### Ethical considerations
Ethical approval was granted by the Swedish Ethical Review Authority (Dnr: 2020–03731). In accordance with the Declaration of Helsinki, the integrity and dignity of the participants have been respected in that their data and interviews were treated confidentially.[26]

### Patient and public involvement
Patients or the public were not involved in the design, conduct, reporting or dissemination plans of this study.

## RESULTS
The nurses perceive advising parents when their child has a fever as four different kinds of balancing acts: balancing between the parents' story and objective assessment, balancing between listening and teaching, balancing between self-confidence and trust in the expert, and balancing between independence and having someone by one's side. All these balancing acts are part of and influence the process of giving advice, but the balancing between independence and having someone by one's side was more dominantly expressed than the others according to table 2. For a schematic view of the outcome space, see figure 1.

### Balancing between the parents' story and objective assessment
The nurses perceive advising on fever as a balance between the parents' story and objective assessment. Not seeing the sick child, as with telenursing, or when the nurse and parent don't speak the same language, is perceived as an additional difficulty. The nurse's objective assessment is based on the parents' subjective interpretation of the child's symptoms and their ability to describe them.

> There, we are also dependent on the parents to actually be able to make an assessment, so they may be dependent on us to get answers, but I am just as dependent on their answers to be able to make my assessment // it's quite interesting actually that it's not just them who need us, but we actually need them, and they know their child best. Q17

The nurses perceived that they are dependent on the parents' stories, but the parents' worries may cause them to assess their children as sicker than they are; at the same time, parents know their children best and their concerns should be taken seriously. This interaction is as a collaboration between nurse and parent where the expertise of the parents concerning their children needs to be respected. The nurses perceive that parents need to provide enough information about the well-being of their children for nurses to be able to make an assessment, which requires good communication skills. Worry is perceived as potentially blocking this knowledge and affect the ability of parents to accurately assess their child's fever. The nurses' perceptions are that the degree of worry can be influenced by several different factors such as number of children, personality, culture, resources and previous experiences. The nurse's perceptions are that some parents have experienced a serious event in connection with a fever; others have read scary stories that are shared in the media and on the internet. Parents from countries where serious infections with fever are more common were perceived by the nurses as having greater worry about fever.

Table 2 Overview of dominant (++) and non-dominant (+) ways of understanding nurses' perceptions of advising parents when their child has a fever

| Interview | The parents' story and objective assessment | Listening and teaching | Self-confidence and trust in the expert | Independence and having someone by one's side |
|---|---|---|---|---|
| 1 | + | + | ++ | ++ |
| 2 | + | ++ | ++ | ++ |
| 3 | + | + | + | ++ |
| 4 | ++ | + | ++ | ++ |
| 5 | ++ | ++ | | + |
| 6 | ++ | ++ | ++ | ++ |
| 7 | + | ++ | + | ++ |
| 8 | ++ | ++ | + | ++ |
| 9 | + | ++ | ++ | ++ |
| 10 | + | + | + | ++ |
| 11 | + | + | + | + |
| 12 | ++ | + | + | ++ |
| 13 | + | + | + | ++ |
| 14 | + | ++ | ++ | ++ |
| 15 | ++ | + | + | ++ |
| 16 | + | + | ++ | ++ |
| 17 | ++ | ++ | + | ++ |
| 18 | + | + | + | ++ |
| 19 | + | + | + | ++ |
| 20 | | + | + | ++ |
| 21 | + | | + | + |
| 22 | + | + | + | ++ |
| 23 | | + | + | + |
| 24 | + | | + | ++ |

The difficult thing is that there is a lot, there can be a lot of worry and then you don't really get the facts, but it is the worries that comes out. J10

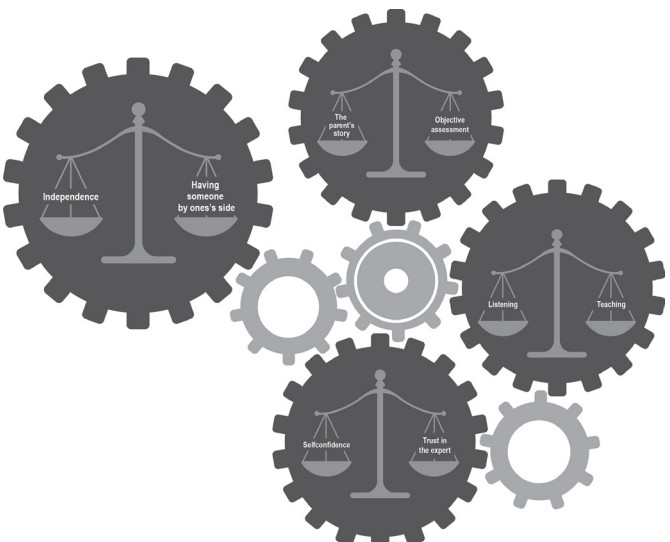

Figure 1 Schematic view of the outcome space.

### Balancing between listening and teaching

The nurses perceive advising on fever as a balance between listening and teaching. Providing information about fever is considered part of a nurse's job, and the interaction between parents and nurses plays a role in how advice is received. It is not just about teaching general knowledge in case of fever, which is something that is simple in and of itself, but the nurse has to listen to the needs of parents and start from there in order to address their specific concerns.

It will be to create a trust and then to make a knowledge inventory and then come along with the information, or the advice that I want to give. If I start by giving advice, I don't think the family sees themselves being seen, then it becomes more dismissive even though it is well-intentioned on my part that I want to help. So it has to happen in the right order there. H8

When time is constrained, advising may become more standardised and less based on individual needs. This is perceived as creating a feeling in the parents that they are not being taken seriously. The nurses also

perceive that sometimes consensus regarding what is classified as a serious illness is lacking, but regardless of what their own assessment is, the parents' concerns need to be confirmed so they feel they are being heard. This includes, for example, booking an appointment to see a doctor even if the nurse assesses this as being unnecessary. At the same time, the nurses perceive that the needs and wishes of the parents must be weighed against the resources of the healthcare system. The nurses' perceptions are that worried parents have the right to seek medical care, but there are benefits to achieving consensus on when the child needs a healthcare visit when it comes to, for example, the waiting time at the paediatric emergency department. In the absence of consensus, the likelihood of parents following the nurse's advice is perceived as reduced. In these cases, it is considered likely that the parents will seek additional advice, either at the same place or elsewhere. Taking the parents seriously, confirming their concerns and explaining the basis for the assessment are perceived as tools for achieving consensus.

> It's important not to diminish their worries but to show that you listen to them and that it's okay that they're worried. But you have to explain to them, if it's the case, that we're not worried and why not in that case, so that they understand our work. E5

### Balancing between self-confidence and trust in the expert

The nurses' perceptions are that advising on fever is a balance between self-confidence and trust in the expert. Nurses at the paediatric emergency department are perceived as having the most expertise advising parents, most often using their own knowledge and local directions as guidance. Nurses in other parts of the healthcare system often described using a CDST. It is considered evidence-based support, mainly for nurses who lack experience, but it is also perceived as requiring some experience to be interpreted correctly. CDST provides a structure and has clear directives when it comes to fever. At the same time, it can be perceived as undifferentiated and limited in more complex cases, and in such cases some nurses place more trust in their own knowledge. The fact that some nurses use CDST and some do not may result in parents receiving different answers depending on where they seek advice. This is perceived as causing confusion and sometimes resulting in contradictions between nurses in different parts of the healthcare system.

> Those who created this decision support, they are specialists, so it's not someone sitting at home and tinkering with this // then, as I said, you have to have your own experience with you, also as a nurse. A newbie, newly trained nurse would not have been so appropriate perhaps to put… in advising, it is not, it is not so simple, so it is not. N14

The nurses perceive though, that the parents generally trust their advice, even if stories about when care has failed to detect and treat serious illness can reduce trust. The trustworthiness is perceived to increase if several people (or sources) say the same thing, if the assessment involves a physician or if the nurse and the parent have an established relation. Establishing trust via phone poses a greater challenge than meeting someone in person. The nurses perceive themselves as providing the most reliable and evidence-based advice and that parents see them as experts and often trust them over themselves.

> There is no opinion in that [the advice], but that it is actually a science that we rely on, I think it is a support that we are actually nurses that they talk to, it's not just anyone, like a grandmother or their mother. T20

Parents are perceived as having different abilities with regard to seeking knowledge about their child's fever, but it is generally perceived as being difficult to sift through all the available information and assess what is reliable. The nurses perceive that this information overload makes it difficult for parents to fully trust their own knowledge or that of their relatives. The nurses also expressed that even if the parents have general knowledge of fever, they may need help to assess the child's condition and require confirmation that their knowledge is being applied correctly to the situation.

### Balancing between independence and having someone by one's side

The nurses perceive that advising on fever is a balance between independence and having someone by one's side. Both training and experience are required, especially on the phone, because fever can represent both harmless and serious conditions. This is also perceived as involving great responsibility as it requires caring for parents as well as the child. The nurse's perceptions are that they need to feel secure in their professional role and not be distressed by the parents' worries. Independently assessing a child with a fever can be perceived as a challenge, especially if it involves a younger child. With increasing experience and knowledge comes independence, but it is still important to have access to support, to have someone by one's side.

> But I think I have the greatest support of my colleagues, so partly I think, despite having worked for a long time, I think it's quite nice that one of my colleagues, it could be the assistant nurse, has also seen the child and says the same as me, that I'm not worried. They have, after all, are often very experienced, have seen a lot of children. Then I feel even more confident in my assessment and can provide better support. B2

This is also perceived by the nurses as being true for the parents. Parents today, though, are perceived as being fearful of fever, of lacking knowledge and support from their relatives, and instead need to seek this support from

the healthcare system. This is by the nurses perceived to be one of the most common reasons for parents to seek advice. Therefore, the nurses perceive that parents need and expect healthcare to be accessible. Providing a plan for how parents should deal with their children having a fever and where to turn when this occurs is perceived by the nurses as a way of increasing the independence of parents, by giving them tools to take control of their situation. This can be combined with a follow-up call, where the nurse can determine whether the parents followed the advice that was given and can evaluate the results. This is perceived by the nurses as them supporting the parents.

> Because then I show that I actually care if the situation has improved or not, if they get the help they need, that someone cares about them and that they are not standing there alone in case something happens. D4

## DISCUSSION

This study understands nurses' perceptions of advising parents when their child has a fever as four different but related balancing acts. Listening to the parents, making assessments and offering advice can be considered the foundation of the advice-giving process, but the nurses' perceptions show that personal characteristics and peer support are important parts of the process. Overall, these results share similarities with those of Greenberg,[27] who has previously described telephone nursing, however, as a process comprising three phases. First phase includes listening to the caller's story and gathering information. In the second phase, the nurse assesses the problem and determines the proper interventions, sometimes with the support of a colleague. The third phase is called output and consists of different nursing actions such as referring to primary care or offering advice for self-management, reassurance and validation. This also describes personal characteristics of the nurse, for example, experience, as influencing factors of the telenursing process.[27]

The nurses in the present study perceive difficulties in balancing between the parents' stories and an objective assessment sometimes as a result of difficulties in communication. Earlier studies show that parents often assess and express symptoms differently than healthcare personnel. For example, parents often start from the view of a child's normal behaviour and judge the health of the child in relation to how much the behaviour deviates from the usual.[28–30] Parents may find it difficult to assess and express specific symptoms such as dehydration and breathing problems if they have not experienced them before.[30] Conversely, healthcare personnel usually assess illness based on specific symptoms, which can lead to mistakes in communication.[31] Greenberg[27] emphasises that nurses interpreting between healthcare information and information that the caller can understand serves as an important component that links together different aspects of the telenursing process.[27] The nurses in the present study perceived that parents' worry affects the way parents assess the symptoms of their children. Previous studies show that it can be difficult for parents to assess their sick child and know when it is time to seek medical care or when self-care is enough.[32 33] Gamst-Jensen *et al*[34] show that in telephone triage there is a relationship between parents' self-rated anxiety and hospitalisation. The perceptions of the nurses in this study suggest that it is important to take parents' concerns into account whether you talk to the parents on the phone or meet them in person.

The nurses in the present study perceive that there had to be a balance between listening and teaching and that their relationship with the parents would decide whether they could reach them with their advice. Halldórsdóttir[18] argues that in order to achieve a caring relationship, the nurse must appear to genuinely care. The nurse needs to build a bridge where the parent feels a sense of belonging in the meeting. If, on the other hand, the parent feels that the nurse is lacking in caring and does not want or cannot meet, a wall is built that makes it difficult to communicate and create trust.[18] For example, the nurses in the present study described a gatekeeping role; even though the nurses wanted to provide service, they felt limited by the resources of the healthcare system. Being refused access to the desired healthcare can give a feeling of not being taken seriously,[29 35] which can then build a wall between the parent and nurse. If there is a disagreement about how serious the child's illness is and the nurse fails to explain her assessment, it can lead to mistrust of the given advice. If the parents are not satisfied with the advice, there is a risk that they will seek advice again in the near future.[36 37] To some extent, this may be due to the parents failing to regain a sense of control over the situation.[28 35]

The nurses in the present study expressed a perceived hierarchy within the healthcare system with regard to who was considered to be experts. A common reason for parents turning to the paediatric emergency department for a minor illness is that they have a higher level of trust in specialist care (often in combination with poor availability in primary care).[14–16 22 38] However, it has also been shown that within the healthcare system there is a large proportion of incorrect referrals to the paediatric emergency department.[14 16 39 40] This could be related to healthcare personnel unwilling to risk making a wrong referral.[41 42] This can strengthen the parents' image that it is in the paediatric emergency department that they will receive the best care and create a search pattern that is inadequate.[43] This can also cause contradictions between the different parts of the healthcare system.[39] It should be added, however, that the majority of studies on telenursing show that telenurses generally advise seeking a lower level of care than the caller first intended.[10 38 44]

The nurses in this study had perceptions about who and what information and advice to trust when caring for a febrile child. While stating that parents could find their

own information, it was understood that this might be difficult for someone without healthcare education, and the nurses perceived themselves to be the experts. Earlier studies have shown that parents are just as aware as the nurses that it can be difficult to know what information is reliable; however, contrary to the perceptions of the nurses in this study, parents do seek, and get, help from both family and friends, as well as the internet, when they need information and advice.[11 33 45 46] On the other hand, when information and advice came from healthcare personnel, it had the highest compliance,[46] which may be a sign that information from healthcare personnel, as well as by parents is considered to be the most reliable information source.

The nurses in the present study perceive a balance between independence and having someone by one's side. Giving advice on fever is perceived as difficult, and the nurses sometimes need support from colleagues in their decision-making when advising parents on this matter, something that has been seen in previously studies as well, however, these studies have focused solely on telenursing.[22 41] Advising via telephone is even more complicated, because the assessment is usually done through a parent. This type of work requires experience, which may be considered even more important than education.[41]

### Strengths and limitations

According to Lincoln and Guba,[47] the trustworthiness of a qualitative study is founded on four pillars: dependability, confirmability, credibility and transferability.[47] In order to ensure dependability and confirmability, the authors have tried to provide a clear description on how the study was performed as well as citations to support the results. None of the authors have any type of relationship with the participants. The first author has experience advising parents when a child has a fever but has never worked in any of the participants' workplaces. Together with the research group, the first author continuously reflected on her own preunderstanding to avoid bias. Interviews were performed, since this may be considered the most common tool to collect data for this kind of study.[24] Since Larsson and Holmström[25] believe that approximately 20 interviews are sufficient to find all the different perceptions that can exist in a group, the 24 interviews that were conducted were considered an appropriate amount of data to handle and they were judged to be rich in different perceptions. Due to the COVID-19 pandemic, the interviews were done via videoconference call, and although most of the participants were accustomed to this way of communicating, it could limit the participants' ways of expressing themselves. Investigator triangulation was used to ensure credibility. Two of the five authors had experience in the phenomenographic method, and all of the authors had experience in qualitative methods. A particularly large focus was on separating what belonged to first-order and second-order perspectives, as this is a fundamental aspect of the phenomenographic analysis. The findings were continuously reflected on until

consensus arose. To ensure that the reader can assess transferability to their own setting, settings and participants were described thoroughly. There may be differences in the educational and healthcare systems of different countries, though, which should be taken into consideration.

## CONCLUSION AND RELEVANCE TO CLINICAL PRACTICE

Giving advice to parents when their child has a fever is a process where the nurse needs to listen, assess and give advice based on the situation. This requires a correct assessment that depends on the parents' story. Creating a trusting relationship is perceived as necessary for parents to assimilate the advice that is provided. What dominates are the nurses' perceptions of the inner qualities required to achieve a balance in the process, for example, the importance of experience and security in their professional role, while it is also necessary to get support from colleagues. This study can be used as a foundation for discussion and reflection for nurses who work with this type of advising, but also as teaching material for students. It can also help nurses from different parts of the healthcare system to understand each other's work. Future research should focus on how interactions between nurses and parents affect the outcome of the advice-giving process from the view of both parents and nurses.

**Author affiliations**
[1]Department of Health and Caring Sciences, Linnaeus University Faculty of Health and Life Sciences, Vaxjo, Sweden
[2]Department of Pediatrics, Region Kronoberg, Vaxjo, Sweden
[3]Department of Caring Sciences, University College of Boras Faculty of Caring Science Work Life and Social Welfare, Boras, Sweden
[4]Department of Ambulance Service, Region Kronoberg, Vaxjo, Sweden
[5]Department of Health and Care, Linkopings universitet, Linkoping, Sweden
[6]Head of Research, Region Kronoberg, Vaxjo, Sweden

**Contributors** All authors (ELW, ILG, AS, MS-L and CE) designed and planned the study together. ELW was responsible for the data collection, conducted and transcribed the interviews. The analysis was mainly performed by ELW, ILG and CE, but all authors participated and discussed the various steps during the process. All authors drafted, edited and finally approved the manuscript. The guarantor for this study is EW.

**Funding** This work was supported by Region Kronoberg grant number 969335.

**Competing interests** None declared.

**Patient and public involvement** Patients and/or the public were not involved in the design, or conduct, or reporting, or dissemination plans of this research.

**Patient consent for publication** Not applicable.

**Ethics approval** This study involves human participants and was approved by the Swedish Ethical Review Authority, dnr: 2020-03731. Participants gave informed consent to participate in the study before taking part.

**Provenance and peer review** Not commissioned; externally peer reviewed.

**Data availability statement** Deidentified data may be available on reasonable request via the corresponding author.

**ORCID iDs**
Emma Westin http://orcid.org/0000-0001-6284-5859
Ingrid L Gustafsson http://orcid.org/0000-0002-5932-6078
Anders Svensson http://orcid.org/0000-0001-7479-8092
Märta Sund-Levander http://orcid.org/0000-0002-1281-885X
Carina Elmqvist http://orcid.org/0000-0001-8376-8805

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
