## [Reviewer comments · BMJ Open]

ARTICLE DETAILS

TITLE (PROVISIONAL)	Advising parents when their child has a fever – a phenomenographic analysis of nurses' perceptions when working at a telephone helpline, at primary care or at a children's emergency department in Sweden
AUTHORS	Westin, Emma; Gustafsson, Ingrid; Svensson, Anders; Sund-Levander, Märta; Elmqvist, Carina

VERSION 1 – REVIEW

REVIEWER	Zilezinski, Max Charité Universitätsmedizin Berlin, Institute of Clinical Nursing Science
REVIEW RETURNED	15-May-2023

GENERAL COMMENTS	Dear Ladies and Gentlemen, Dear authors, First of all, I would like to thank you for the opportunity to review your article. I find your work interesting, important, and relevant to nursing. To give you the impression that I understand your work, I would like to provide a brief summary of my comments and then I will explain them in more detail. Please consider my comments as recommendations. This is foundational work that, as I understand it, builds on previous work of yours. This work makes an important contribution to understanding the perceptions of nurses' when advising the parents of children have fevers. Therefore, you have selected different types of participants (nurs-es working with parents and children with fever in different settings/workplaces) to provide a broader overview (understanding) of the process of advising parents in a (perhaps) unfamiliar or difficult situation (child with fever). Based on the findings, the manuscript offers relevant insights into the process of advice given to parents by nurses. It makes an important contribution to the understanding of the process from the nurse's perspective and thus offers the possibility to train and support nurses in this role. My specific comments are based on both the reporting statement and your manuscript, and on the following points: Transferability, Credibility/ Confirmability, Reflexivity, Transparency/ Dependability. In this regard, I used the following article: Williams V, Boylan A, Nunan D: Critical appraisal of qualitative research: necessity, partialities, and the issue of bias. BMJ Evidence-Based Medicine 2020; 25:9-11. Please see the attached document.
---

REVIEWER	Flenady, Tracy Central Queensland University, School of Nursing, Midwifery and Social Science
REVIEW RETURNED	06-Jun-2023
GENERAL COMMENTS	Thank you for the opportunity to review this worthwhile manuscript. the topic is interesting and examines a phenomenon that most nurses experience in their day to day role.

VERSION 1 – AUTHOR RESPONSE

Return of the review to the editorial office: 19.05.2023

General comments/ summary comments on the manuscript	Authors answer
First of all, I would like to thank you for the opportunity to review your article. I find your work interesting, important, and relevant to nursing. To give you the impression that I understand your work, I would like to provide a brief summary of my comments and then I will explain them in more detail. Please consider my comments as recommendations. This is foundational work that, as I understand it, builds on previous work of yours. This work makes an important contribution to understanding the perceptions of nurses' when advising the parents of children have fevers. Therefore, you have selected different types of participants (nurses working with parents and children with fever in different settings/workplaces) to provide a broader overview (understanding) of the process of advising parents in a (perhaps) unfamiliar or difficult situation (child with fever). Based on the findings, the manuscript offers relevant insights into the process of advice given to parents by nurses. It makes an important contribution to the understanding of the process from the nurse's	Dear Mr Zilezinski Thank you very much for taking the time to review our script. You raise many good points and these have been a valuable help in improving our text. We have inserted our answers to your comments below.

perspective and thus offers the possibility to train and support nurses in this role. My specific comments are based on both the reporting statement and your manuscript, and on the following points: Transferability, Credibility/Confirmability, Reflexivity, Transparency/Dependability. In this regard, I used the following article: Williams V, Boylan A, Nunan D: Critical appraisal of qualitative research: necessity, partialities, and the issue of bias. BMJ Evidence-Based Medicine 2020; 25:9-11.	
--	--

Chapter / Section	Lines/ parts/ written words	Comments, notes and recommendation of the reviewer	Authors answers
Title	Advising parents when their child has a fever – a phenomenographic analysis of nurse's perceptions	Your title makes it clear what you have done, so that the reader can immediately see what you are talking about. I have no comments on your title.	Thank you, the title is changed on request from the editor.
Abstract	In general	The abstract gives us the opportunity to get an overview about the article, but it provides the opportunity for streamlining and compression as specially the section design, setting and participants.	Yes you are right, see our answers below.

	Line 16 – 17 (design)	I know that the authors' guideline suggests that in the abstract "design" must be written. In contrast, the reporting guideline suggest using method for describing your study. If it is possible, I would choose "method" and streamline your written text. For example: Design/method: Inductive, descriptive study with a qualitative, phenomenographic approach. I am not sure that the wording "descriptive" is correct for your research approach. If I understand it correctly, there are no previous studies on your topic and your work is fundamental to this topic. Therefore, I would understand your work as exploratory and change the wording.	Thank you for suggesting that, if it is ok with the editor we will make that change, see yellow markings. About the word "descriptive", that is the correct term for a phenomenographic approach.
	Line 20 – 33 (setting / participants) In general	I know the author's guideline suggests a different order, but for the sake of readability, I suggest changing the order or making the sections (Setting/ Participants) to one (Participants and setting), if possible. Than you have the option to streamline your text. For example: Participants and setting: A strategic sample was used. To be included, the 24 online interviewed nurses had to have experience advising parents of febrile children between birth and five years of age. They were recruited from three different parts of the healthcare system from four regions in the south of Sweden.	That is a really good proposal, if it is ok with the editor we will make that change, see yellow markings.

	Line 25 (participants)	At this point, the participants are unclear, I would that you talk about nurses at this point.	Yes, that is correct.
	Keywords	I would suggest checking if your keywords are already in your title or abstract and if they are mesh terms or not. Maybe you can use “Mesh on demand” to find possible mesh terms as keywords for your abstract. (https://meshb.nlm.nih.gov/MeSHonDemand)	Thank you for your suggestion regarding keywords. In the online submission system, there were only a few that matched our work. We probably would have chosen others if the opportunity had existed.
Strengths and limitations of this study	Line 48 et seqq.	Gives the reader an essential overview and is, in my opinion, sufficient in its execution. The section is important for your credibility/confirmability and transparency/dependability.	Thank you.
Introduction	In general	The introduction is sufficient in its length, it takes the reader along and contains relevant information on the topic, and moreover, it is theory-based. Nevertheless, I would like to give you some suggestions, especially around insights into the counseling process between nurses and parents can be elaborated more clearly.	Thank you, see our responses below

	Line 15 – 23	Here I suggest it is worth taking another look at the process of consultation between nurses and parents in general. In addition, I would recommend that you outline key points from the studies on parents to support the research gap you describe. Why is research needed at all - the goal remains unclear!	We have chosen to move up the section on the definition of advising to make it clear that it is these parts that the nurses perform, see yellow markings line 3-5. Have also tried to clarify why this study is needed, line 31-34.
	Line 25 – 55	To my mind, there is a break in the reading between the third and your fourth paragraph of the introduction. You may be able to resolve this by making the third paragraph the first paragraph of your introduction and then outlining the specifics of your topic.	During the process of writing this manuscript there have been a lot of thoughts about which order is the most preferably. We have now changed it according to your advice and feel that this creates a better flow, thank you for that.
	Line 46 (references 19/20)	The reference you listed (19) is a review paper, so as I understand it, citing one source seems sufficient to me. Even if Reference 20 was not part of the review paper. I would avoid self-citation, if possible, please check critically if both sources are necessary.	The review paper doesn't cover fever seizures as the other paper does. We do feel that that contributes to the whole picture.

	Line 50 (references 21)	I have not been able to trace the reference to verify your statement, because the reference is only in Swedish. Are the reference necessary at this part of the manuscript?	Thank you for the point of view, we think this describes well what is meant by advising in a Swedish context but have moved it up a bit in the manuscript (line 3-5).
--	----------------------------	---	---

Methods	In general Line 5 (Design)	As I already mentioned at the beginning of the abstract, it is important to check to what extent the description of the design (in the method section) is required by this journal. In addition, it is important to critically examine to what extent parts of this section do not belong to the method but to the results (e.g. the tables). What do you mean by inductive? Your summary of the interview procedure is a good start, though I believe there might be some room for further details. Specifically, it would be helpful to understand more clearly the semi-structured nature of the interviews. Could you perhaps shed some light on the questions that were guided by your interview protocol? It would also be interesting to know if there were multiple questions posed to the participants. Including the interview guide could be a great way to enhance the readers' understanding. Furthermore, it would be beneficial to elaborate on the impact that the pilot interviews had on shaping the questions. Such information would undoubtedly contribute to a more comprehensive understanding of your research process	There is no detailed description from the journal of what should be included under the heading design. However, it feels important to give a brief description of how phenomenography differs from other approaches, as it is written in the script now. Regarding the tables, see answer below. Inductive means that the analysis has not been done with predetermined concepts or a template based on theory. We will attach the interview guide. Thank you for pointing that out, actually, the questions did not
---------	--	---	---

			change after the pilot interviews, we have now clarified that in the script, see yellow markings line 11-12.
	Participants	You address the different parts of the health care system in the abstract. The section on participants lacks a description of this. I would recommend that you comment briefly on this in the text and do without a description in the abstract. How did you get the 24 participants? The recruitment process is not comprehensible on the basis of the explanations in the manuscript; here I recommend that you expand the text with important points as part of your methodological explanations. Were the 24 participants necessary, as data saturation then occurred?	Yes, we have now moved the part about the health care system from the abstract to this part of the script and clarified why we included 24 participants, see yellow markings line 20-24. That really improved that part, thank you.

Participants Line 18 (strategic sample)	The term "strategic sample" is not as clear as it could be. If I understood your sampling strategy correctly, it was based on predefined criteria (different parts of the health care system and from four different regions in southern Sweden, and experience in advising parents). Then it is my understanding that this is a criterion sampling, which is a specific form of purposive sampling. I would suggest you a critical examination of the terminology. My comment is based on the explanations in Polit & Beck (2021): Nursing Research. Generating and Assessing Evidence for Nursing Practice. 11Ed. (International Edition). Wolters Kluwer	Yes! Thanks for the suggestion, have changed to purposive which is also common in phenomenographic method, see line 11.
Table 1 – Title	In my opinion, there is potential for shortening the title.	Yes, we have done that, see line 29
Table 1	They address the different area of the health care system in the abstract. The section on participants lacks a description of this. I would recommend that you comment briefly on this in the text and do without a description in the abstract.	Yes, we have changed that, see comment above.
Table 1	The line heading "1177" is not self-explanatory or comprehensible to the reader at first glance, so I would recommend that you replace it with "telephone helpline" or "nurse telephone triage".	Yes, we have changed that, see table 1.
Table 1	There is still capacity to expand on the description of the participants at this stage, so I would recommend you to describe the sample in more detail. For example, the specialization can be explained in more detail in the table, as well as the work experience.	We have developed table 1, however it is not appropriate to print more details about the specialist nurse educations as it risks confidentiality.

	Table 1	Even though the range of work experience is already given in the text, it would be interesting to know if there was a difference between the different groups. It makes sense to insert the range in the mean work experience.	See above
	Data analysis Line 19 - 25	In the Reporting Statement, I noticed that the process of data analysis could benefit from a more comprehensive description. Upon reviewing the text multiple times, I found it challenging to identify who was in charge of the process and which researchers were involved. It would also be helpful to know if there was an internal review process involving multiple people. I kindly suggest taking a closer look at this section and considering a revision to provide greater clarity. Additionally, it would be valuable if you could clarify your approach to handling and processing the data. Understanding the details of the process would contribute to a more transparent and thorough presentation.	Thank you. We have clarified the various steps in the data analysis and described who did what, see yellow markings line 21-15 and box 1. Under data collection (see yellow markings line 13-16) there is now more information about how the data was handled and processed according to the reporting statement.
	Table 2 - Title	In my opinion, there is potential for shortening the title.	Thank you for sharing your opinion. We have looked that over but feel that this is the way we want to name this table.

	Table 2	The content you presented in table two is, in my opinion, should be included in the presentation of the results. Please check to what extent the table can also be accommodated in the results section.	Thank you for this suggestion, but this type of table is usually placed under the analysis section in phenomenographic approach.
	Ethical considerations	You cite the same reference twice in the same sentence. In think it is enough to indicate the reference at the end. I would recommend that you delete the first one (25).	Good that you noticed that, it is now removed.

Results	In general	The description of the results already anticipates part of the discussion in various passages. I recommend a clearer differentiation of the individual sections (results and discussion) – I think some work on internal consistency and structure would be helpful and necessary. In addition, I am surprised that within the framework of phenomenology there is already a quantification of the results at this stage (Figure 1). Explaining how you came to this conclusion would make things clearer to the reader. The category Independence is described vaguely - it is necessary to sharpen the description and to check whether the key citations are really suitable. In addition, the addressees are mixed up.	We have tried to clarify in some places that it is the nurses' perceptions that are being written about, to make it clearer that it is a result and not a discussion, see yellow markings. Within the phenomenographic method, it is usual to visualize the outcome space with a figure in the way we have done. But we hope that the additional description of the analysis in box 1 as well as some clarifying in the first paragraph under results will help make the figure clearer. What emerges under the category of independence is what emerges in the analysis of the interviews and the quotes describe
---------	-------------------	---	---

			the nurses' perceptions.
	quotes	Your given quotes are comprehensible and plausible. For me in the result execution also sufficient. Perhaps there is the possibility of further citations to the respective topics in an appendix. Especially against the background of your many interviews.	Thank you, we do also believe that the quotes are sufficient but if the editor allows additional quotes in an appendix we will consider this.
	figure	I have not found a subtitle to the illustration. At the same time, check whether the figure is still necessary to better convey the content to the reader.	The illustration is figure 1.

Discussion	Yes	The discussion of the results is fundamentally interesting and supported by empirical evidence. It would be helpful for the readers if you briefly mentioned the listed literature in the introduction and then referred to it in the discussion. If I understood you correctly, your results are in part comparable to other empirical findings. This comment refers to a comment on the introduction (the second comment).	Thank you! The introduction provides a background and is structured based on purpose, organization and conditions, not according to what the results show, thus some references in the introduction are not included in the results. It is true that in some parts our results agree with other studies, however, it is often only parts of the context that are the same, for example many previous studies were done in telenursing and not in several parts of the healthcare system as this study is. We have tried to clarify this in the discussion, see yellow markings.
------------	-----	---	---

	Line 10 -16	The category does not exist in the presentation of results, where does this come from instead of this category? (...) but the nurses' perceptions show that this is less dominant than the category containing personal characteristics and support. (...)	yes this was unclearly written and could be misinterpreted, have now reformulated that sentence, see yellow markings line 31-32.
Strengths and limitations	In general	I think with the explanations to strengths and limitations should also consider what extent contents or supplementing explanations in yours methods took place. Especially with regard to the "Techniques to enhance trustworthiness" mentioned in the Reporting Statement.	Thank you for the comment, unfortunately we don't quite understand what you mean. From our point of view, we have described all the concepts you mention based on Lincoln and Guba (1985). You are welcome to clarify what you mean.
Conclusion and relevance to clinical practice	In general	The conclusion is extensive; the extent to which there is potential for cuts would have to be clarified. In addition, redundancies for discussion should be avoided here as far as possible.	Yes, that is a valid point! Hope the changes we have made in conclusions have corrected that.
References	In general	The references are sufficient in their execution. I could not check some of the references due to the language barrier.	Thank you.

Reporting-Guideline Check	In general	Thank you for completing and attaching the reporting guideline. It was helpful for us as reviewers.	Thank you for a very thorough and helpful review!
------------	---	---

VERSION 2 – REVIEW

REVIEWER	Zilezinski, Max Charité Universitätsmedizin Berlin, Institute of Clinical Nursing Science
REVIEW RETURNED	22-Dec-2023

GENERAL COMMENTS	Thank you very much for revising your manuscript and taking my comments into account. I congratulate you on a successful publication. Merry Christmas and a Happy New Year.
---